# Non-Surgical Transversal Dentoalveolar Compensation with Completely Customized Lingual Appliances versus Surgically Assisted Rapid Palatal Expansion in Adults—The Amount of Posterior Crossbite Correction

**DOI:** 10.3390/jpm12111893

**Published:** 2022-11-11

**Authors:** Jonas Q. Schmid, Elena Gerberding, Ariane Hohoff, Johannes Kleinheinz, Thomas Stamm, Claudius Middelberg

**Affiliations:** 1Department of Orthodontics, University of Münster, 48149 Münster, Germany; 2Department of Orthodontics, Hannover Medical School (MHH), 30625 Hannover, Germany; 3Private Practice, 49152 Bad Essen, Germany; 4Department of Cranio-Maxillofacial Surgery, University of Münster, 48149 Münster, Germany

**Keywords:** crossbite, surgically assisted rapid palatal expansion, surgically assisted rapid maxillary expansion, dentoalveolar compensation, expansion, mandibular constriction, mandibular compression, lingual orthodontics

## Abstract

The aim of this study was to compare the crossbite correction of a group (*n* = 43; *f*/*m* 19/24; mean age 27.6 ± 9.5 years) with surgically assisted rapid palatal expansion (SARPE) versus a non-surgical transversal dentoalveolar compensation (DC) group (*n* = 38; *f*/*m* 25/13; mean age 30.4 ± 12.9 years) with completely customized lingual appliances (CCLA). Arch width was measured on digital models at the canines (C), second premolars (P2), first molars (M1) and second molars (M2). Measurements were obtained before treatment (T_0_) and at the end of lingual treatment (T_1_) or after orthodontic alignment prior to a second surgical intervention for three-dimensional bite correction. There was no statistically significant difference (*p* > 0.05) in the amount of total crossbite correction between the SARPE and DC-CCLA group at C, P2, M1 and M2. Maxillary expansion was greater in the SARPE group and mandibular compression was greater in the DC-CCLA group. Crossbite correction in the DC-CCLA group was mainly a combination of maxillary expansion and mandibular compression. Dentoalveolar compensation with CCLAs as a combination of maxillary expansion and mandibular compression seems to be a clinically effective procedure to correct a transverse maxillo-mandibular discrepancy without the need for surgical assistance.

## 1. Introduction

Posterior crossbite is a common malocclusion with a global prevalence of 10% in the permanent dentition and a prevalence of up to 15% in the European population [1,2]. The etiology of posterior crossbites remains unclear [3] but they may be associated with non-nutritive sucking habits [4] or pacifier use [5]. Breastfeeding seems to be a protective factor against the development of posterior crossbites [6].

Treatment options for posterior crossbite usually include different kinds of maxillary expansion [3] whereas a change of the mandibular archform is less common due to concerns about instability [7]. Maxillary arch expansion in the early mixed dentition can be done quite easily with removable plates [8]. The interdigitation of the midpalatal suture increases with age [9] and greater forces may be necessary for maxillary expansion in the early permanent dentition, which is usually done by either fixed tooth-borne, tooth–bone-borne or bone-borne maxillary expansion devices [3]. Problems arise in the skeletally more mature patients in late adolescence or early adulthood, when conservative maxillary expansion is contraindicated because of changes in the osseous articulations of the maxilla with the surrounding bones [10,11]. Many attempts have been made to assess the mid-palatal suture maturation [12,13] or suture density to facilitate the decision between conservative or surgically assisted techniques for maxillary expansion, but the evidence is weak and it remains a subjective decision to opt for surgically assisted rapid palatal expansion (SARPE) in late adolescents and early adults [14], because the midpalatal suture is not the only resistance to conservative maxillary expansion and the circummaxillary sutures have to be considered as well [11].

To date there is no consensus about either the extent or procedure for SARPE [10,15] or the use of a bone-borne or tooth-borne expander in these cases [16,17]. Another treatment option is microimplant-assisted rapid palatal expansion (MARPE) using tooth–bone-borne expanders. There is evidence that MARPE can increase the success rate in separation of the midpalatal suture [18] and leads to greater skeletal effects [19] in adolescents after the growth spurt. MARPE also shows a parallel expansion pattern in young adults [20,21].

Complications with SARPE, such as asymmetrical expansion in 5% of the cases, remain a problem [22]. Skeletal and dental effects of SARPE are controversial in the literature. A prospective study by Asscherickx et al. in 2016 found mainly skeletal effects [23], while a recent systematic review of randomized clinical trials showed primarily a molar expansion rather than a bodily skeletal expansion of the maxilla [24].

Segmental osteotomy or two-piece maxilla has the advantage to save one operation. But it is suitable only for mild transversal discrepancies and up to 60% of the gained expansion relapses [25].

This draws attention to dentoalveolar compensation as a possible therapy option for adults. Some authors suggest that transversal discrepancies of up to five millimeters can be corrected by orthodontic tooth movement only [10,26]. Two things are needed for this tooth movement: torque control to reduce tipping, and an adequate force. This can be accomplished by the use of a completely customized lingual appliance (CCLA), which can accurately achieve the planned tooth position of the initial setup [27,28,29], has a good torque control due to the high precision of the slot–archwire combination [30,31] and offers biomechanical advantages for expansion and compression due to the shorter interbracket distance [32], and therefore a shorter total archwire length.

The purpose of this study was to compare the amount of crossbite correction by SARPE versus non-surgical dentoalveolar compensation (DC) using CCLA in adults with posterior crossbite. The null hypothesis was tested that total crossbite correction with SARPE is greater than with dentoalveolar compensation.

## 2. Materials and Methods

This retrospective cohort study was approved by the local Ethics Commission of the Medical Faculty of the University of Münster, Germany (2021-120-f-S). The study protocol was described according to the STROBE Guidelines [33]. The measurements took place at the University Hospital Münster, Germany. To compare the amount of crossbite correction two groups were formed: a surgical group treated with SARPE followed by a buccal straight wire appliance, and a DC-CCLA group treated with the WIN appliance (DW-Lingual Systems GmbH, Bad Essen, Germany).

Inclusion criteria were adult class I, II or III patients with crossbite of two or more teeth in the posterior segments. Exclusion criteria were patients with syndromes, clefts, primary failure of eruption and genetic tooth agenesis, such as hypodontia or oligodontia with and without associated systemic disorders. Teeth that were moved in the sagittal direction for space closure because of extractions or space opening for prosthetic replacement of a tooth were excluded from the measurement.

The DC-CCLA group consisted of consecutively debonded patients treated in a private practice (Bad Essen, Germany) during the period from 2019 to 2021. The transversal dimension was measured on digital models derived from plaster casts before treatment (T_0_) and after debonding (T_1_). All patients in this group were treated with individual archwires manufactured by a bending robot [34]. To obtain the necessary transversal correction, an extra expansion of 1, 2 or 3 cm was incorporated in the maxillary 0.016 × 0.024-inch stainless steel archwire depending on the actual clinical situation. This represents a slow expansion of the maxillary arch. For the correction in the mandibular arch, 1 or 2 cm of compression was incorporated in the same type of wire by the robot. The corresponding bends were added in the anterior segment from 3-3 in each interbracket distance. No cross elastics were used for the expansion of the upper or constriction of the lower arch.

The SARPE group consisted of adult patients who consecutively underwent surgery at the Department of Cranio-Maxillofacial Surgery, University Hospital Münster, Germany in the period from 2018 to 2021. The respective measurements were made on digital models derived from plaster casts before treatment (T_0_), and intraoral scans after bignathic alignment prior to a second surgical intervention for three-dimensional bite correction (T_1_). The surgical concept at the University Hospital Münster routinely involves a two-stage procedure when surgically assisted expansion is needed. First, a SARPE is performed. The surgical procedure for SARPE consists of a subtotal Le-Fort I osteotomy with a separation of the pterygomaxillary junction. The decision for a bone-borne or tooth-borne appliance is made by the referring orthodontist on a case-by-case basis. Bonding of orthodontic brackets is performed six months after the surgical intervention to initiate the subsequent full orthodontic preparation. This is generally followed by a single jaw bilateral sagittal split osteotomy (BSSO) or a bignathic intervention in combination with a Le-Fort I osteotomy. A precondition for the second stage surgery is that the dental arches of the upper and lower jaws are levelled, aligned and coordinated in the transverse dimension with a passive rectangular stainless steel archwire in place. The Digital Münster Model Surgery (DMMS) system for planning is published elsewhere [35].

The plaster casts were scanned with the ATOS II system (GOM, Braunschweig, Germany) and intraoral scans were taken with TRIOS 3 (3Shape, Copenhagen, Denmark). Stereolithography (STL) files were exported to Meshmixer (Autodesk, Inc., San Rafael, CA, USA), with which all registrations and measurements were performed.

### 2.1. Measurement Process

The first step in the transversal measurement was the alignment of the pretreatment (T_0_) STL files to Meshmixer’s world frame. The world frame is the canonical axis-aligned coordinate system, where the *y*-axis always points upwards, and the *x*-axis always points to the right. The maxilla was symmetrically oriented along the palatal suture to the midsagittal plane and with its occlusal plane parallel to the world frame’s ground-plane grid (Figure 1). Possible tilts of the occlusal plane around the *z*-axis (anteroposterior) were adjusted.

The STLs at T_1_ were manually preregistered to the STLs at T_0_. After that, the T_0_ file was selected as a target object and the palatal surface of the T_1_ STL was selected with the brush selection tool. In edit mode, the selected surface was then aligned to the target object with an error tolerance of 0.01 mm (Figure 2). After registration, the digital models were checked again for symmetrical alignment.

To measure strictly in the transverse plane, a surface object parallel to the median plane was placed in the first landmark of one side. The transform tool was then used to shift the surface object along the *x*-axis to the corresponding measuring point of the other side. In the transform window, the translation distance was then taken, which corresponds to the distance between the two landmarks in the mediolateral direction (Figure 3).

A similar procedure was followed with the mandible, with the exception that the initial alignment in Meshmixer’s world frame was based on the occlusion at T_0_. The following landmarks were used for transverse measurements: the tip of the upper and lower canines (C), buccal cusp tips of upper and lower second premolars (P2) and the tip of the mesio-buccal cusp of the upper and lower 1st and 2nd molars (M1, M2). In some cases, the predefined dental landmarks had to be altered because abrasion or restorations hindered reproducibility. Then, other more reproducible surface structures were determined on the teeth.

Maxillary correction, mandibular correction and total crossbite correction were calculated from these measurements for both groups. To calculate total crossbite correction, values for the maxilla and mandible were added or subtracted accordingly (maxillary expansion + mandibular compression or maxillary expansion − mandibular expansion).

### 2.2. Statistical Analysis

To measure the relation between the intervention (SARPE, DC-CCLA) and gender as well as intervention and Angle class, a Chi-squared test was used. To determine if there were differences in transversal changes between the SARPE and DC-CCLA group, independent *t*-tests were used. Equality of variances was tested using Levene’s test. The Welch test was used if equal variances were not assumed. The significance level was set to α = 5% and a *p*-value *p* < 0.05 was considered significant. No α-correction for multiple testing was performed due to the exploratory nature of the study. All statistics were performed using the software SPSS Statistics 27 for Mac (IBM Corp., Armonk, NY, USA).

The method error was assessed by intraindividual reproducibility of the measured distances. For this purpose, the principal investigator (JQS) measured ten randomly selected models at two different time points. The measurement error was determined using Dahlberg’s formula [36].

## 3. Results

The SARPE group consisted of 43 patients (mean age 27.6 ± 9.5 years): 19 females (mean age 26.7 ± 9.7 years) and 24 males (mean age 27.8 ± 9.8 years). The DC-CCLA group included 38 patients (mean age 30.4 ± 12.9 years): 25 females (mean age 32.1 ± 12.3 years) and 13 males (mean age 27.0 ± 14.2 years) (Table 1). No relation was found between gender and intervention (SARPE, DC-CCLA), but there was one found for intervention and Angle classification (Chi-square 32.23, *p* = 0.000, Cramers V = 0.64). Therefore, a split analysis was not performed. According to Dahlberg’s formula, a measurement error of 0.36 mm must be assumed for this study.

The two groups did not differ in age at the start of treatment (T_0_, *p* > 0.05) or at the end of treatment (T_1_, *p* > 0.05) (Table 1).

### 3.1. Maxillary Correction

Maxillary correction showed a typical pattern in both the SARPE and DC-CCLA groups (Figure 4). Within the SARPE group, mean expansion was lowest in the canine region (2.3 ± 3.0 mm) and greatest at the first molars (5.7 ± 2.6 mm). The mean dental expansion at P2 was 5.4 ± 3.0 mm and at M2 4.7 ± 2.8 mm (Table 2).

In the DC-CCLA group, mean expansion was lowest in the second molar region (0.4 ± 2.6 mm) and greatest at the second premolars (4.1 ± 2.9 mm). The mean dental expansion at C was 1.8 ± 1.9 mm and at M1 3.7 ± 2.5 mm (Table 2).

Overall, there was a statistically significant difference in mean maxillary expansion between the SARPE and DC-CCLA groups at P2 (*p* < 0.05), M1 (*p* < 0.001) and M2 (*p* < 0.001). There was no statistically significant difference between the groups at the canines (*p* > 0.05).

### 3.2. Mandibular Correction

In the SARPE group, mean arch width decreased by −0.4 ± 1.7 mm at the canines and −0.4 ± 1.5 mm at the first molars. The mean width increased by 0.4 ± 2.3 mm at P2 and by 0.9 ± 1.2 mm at M2 (Figure 5, Table 2).

In the DC-CCLA group, there was a mean expansion of 0.2 ± 1.7 mm at the canines and a constriction that increased from P2 to M2. Mean arch width decreased by −0.7 ± 2.5 mm at P2, −2.7 ± 2.3 mm at M1 and −3.5 ± 2.6 mm for M2 (Table 2).

There was a statistically significant difference in mean mandibular correction between the SARPE and DC-CCLA groups at M1 (*p* < 0.001) and M2 (*p* < 0.001). There was no statistically significant difference between the groups at the canines (*p* > 0.05) and at the second premolars (*p* > 0.05).

### 3.3. Total Crossbite Correction

For the entire correction, the values for the maxilla and mandible were added or subtracted accordingly (Figure 6). Within the SARPE group, mean total crossbite correction was lowest in the canine region (2.2 ± 2.6 mm) and greatest at the first molars (5.8 ± 2.4 mm). Total crossbite correction at P2 was 4.8 ± 2.7 mm and at M2 3.3 ± 3.1 mm (Table 2).

In the DC-CCLA group, mean total crossbite correction was also lowest in the canine region (1.3 ± 2.1 mm) and greatest at the first molars (6.0 ± 2.8 mm). Total crossbite correction at P2 was 4.4 ± 2.7 mm and at M2 4.1 ± 2.9 mm (Table 2).

There was no statistically significant difference in total crossbite correction between the SARPE and DC-CCLA groups at C, P2, M1 and M2 (*p* > 0.05). There were similar maximum values at P2 (9.1 vs. 9.3 mm), M1 (10.1 vs. 11.2 mm) and M2 (9.7 vs. 9.4 mm) in both groups (Table 2).

When comparing the two appliances for SARPE, there was no statistically significant difference (*p* > 0.05) in total crossbite correction between tooth-borne and bone-borne appliances (Figure 7).

## 4. Discussion

The aim of this study was to compare the amount of crossbite correction between a SARPE and a DC-CCLA group. This was done in terms of transverse metric distance and not in terms of torque measurements. Based on the results of this study, no substantial difference in total crossbite correction could be detected. The null hypothesis that net transversal changes with SARPE are greater than with dentoalveolar correction must be rejected.

Specifically, the SARPE group shows a greater expansion of the upper arch, whereas in the DC-CCLA group a significant portion of the transverse correction is due to the constriction of the lower arch. A clinical example illustrating DC is shown in Figure 8.

Many treatment options for posterior crossbite have been described in the literature but there is a lack of studies that involve a constriction of the lower arch. A recent systematic review revealed a mean expansion at M1 of 7.0 mm (95% CI, 6.1–7.8) immediately after SARPE [24]. The fact that our values are slightly smaller (mean = 5.7 mm; 95% CI, 4.8–6.6) can be explained by the fact that the measurements were not taken directly after surgical expansion, but only after the arch form had been prepared for subsequent BSSO and/or Le-Fort I osteotomy. This is in accordance with the results of Chamberland and Proffit [37] who found a mean expansion at M1 of 6.6 mm (95% CI, 5.2–6.8) at the time of the second surgical procedure.

Our results confirm the opinion that transverse discrepancies of five millimeters can be corrected by orthodontic tooth movement only [10,26]. Looking at the maximum total crossbite correction achieved, it is noticeable that in both groups the values at P2 (9.1 vs. 9.3 mm), M1 (10.1 vs. 11.2 mm) and M2 (9.7 vs. 9.4 mm) were quite similar (Table 2). On the other hand, some expansion values were also within the measurement error.

### Strength and Limitations of the Study

The present study focused exclusively on the comparison of surgical and dentoalveolar crossbite correction. It was necessary to find two comparable situations and we assumed this to be the case in the establishment of the arch form seen at the end of treatment for the DC-CCLA group and at the end of orthodontic preparation for the second surgical procedure for the SARPE group. Although the palates differ before and after treatment, best-fit alignment of the palatal surface was considered an objective and reproducible method to reduce asymmetric measurements between landmarks. In general, measurements of arch width with manually set landmarks on digital models can be considered accurate and reliable [38,39]. However, the study has limitations that must be considered when interpreting the results.

Due to the retrospective design of the study the surgical patients were treated by various referring orthodontists. It can therefore be assumed that different arch forms were used, which naturally have an influence on the result of maxillary expansion. In addition, the decision to use tooth-borne or bone-borne appliances was made by the orthodontist on a case-by-case basis, contributing to the heterogeneity of the SARPE group. A further influence is the selective positioning of bone-borne distractors by the surgeon [40] resulting in more anterior, posterior or asymmetric expansion. Additionally, the separation of the pterygomaxillary junction [41,42], which is the standard at the Department of Cranio-Maxillofacial Surgery, makes it difficult to compare the results to other studies that do not perform this disjunction [16] or use a complete Le-Fort I osteotomy with segmentation of the maxilla [43] and thus have a different pattern of expansion. However, this separation seems to be important to prevent complications at the cranial base [42,44].

In the non-surgical group, individual archwires were used. These were manufactured by a bending robot [34] to incorporate an extra expansion of 1, 2 or 3 cm in the upper arch or compression of 1 or 2 cm in the lower arch. This was done to fully achieve the planned transversal change of the setup because it was shown that CAD/CAM lingual appliances without extra-expansion could not fully achieve the planned expansion at the second molar without these expansion wires [29]. The amount of extra-expansion was determined on an individual basis, which can be seen as another limitation of this investigation. The corresponding bends were added only in the anterior segment from 3-3 to avoid severe changes of the arch form.

The two groups differ in terms of Angle classification. This was to be expected, since two different protocols for correcting malocclusions were studied here. The surgical group includes surgical sagittal correction in every case, which explains the higher proportion of class II and III cases. In addition, the demand of transverse correction is greater for a class II than for a class III case, which must be considered when interpreting the results of both groups.

Despite all the above-mentioned influences, which mainly concern the surgical procedure of transverse expansion, it must be noted that the dentoalveolar compensation achieved comparable, strictly speaking not significantly different results. Additionally, this is not about the size of the possible maxillary expansion, but solely about the correction of the crossbite. Therefore, factors such as stability and side effects of both treatment modalities must also be taken into account when assessing the success of treatment [45].

The design of this study does not allow conclusions about stability. Most authors agree that expansion in the lower jaw is contraindicated because of the high risk for relapse and that the pretreatment arch form serves as a reference and should be retained [7,46], but little is known about the effects of constriction in the lower arch since arch width seems to inevitably decrease over time [47]. It should be emphasized that not every crossbite has its only cause in a narrow upper arch. In these cases, a modification even of the mandibular arch seems justifiable as long as biological limits are respected.

Complications of SARPE, such as epistaxis, asymmetric expansion, periodontal problems or postoperative pain, remain a problem. A recent systematic review [22] found an 22% incidence of overall complications from SARPE and these complications seem to vary significantly across studies [48]. In addition, according to the literature, SARPE shows also a significant amount of tipping of the maxillary halves [23,49] and not a parallel bodily skeletal expansion of the maxilla [24]. Although patient satisfaction with SARPE appears to be high [50] and it can be considered a well-established treatment modality for many years, it is medically reasonable to consider dentoalveolar compensation in borderline cases [51].

The present study does not simplify the decision for surgical or dentoalveolar correction of the crossbite in adults. Prospective studies are necessary to clarify the open questions about long-term stability and possible side effects such as periodontal problems of both treatment protocols.

## 5. Conclusions

There was no statistically significant difference in total crossbite correction between SARPE and dentoalveolar compensation in terms of transverse metric distance. Long-term stability or the extent of angular inclination cannot be extrapolated from the study results. SARPE leads to a greater expansion of the upper arch, whereas dentoalveolar correction with fully customized lingual appliances leads to a greater constriction of the lower arch. Considering the limitations of this study, dentoalveolar compensation with CCLAs seems to be a clinically effective procedure to correct a transverse maxillo-mandibular discrepancy without the need for surgery when constriction of the lower arch is justifiable.

## Figures and Tables

**Figure 1 jpm-12-01893-f001:**
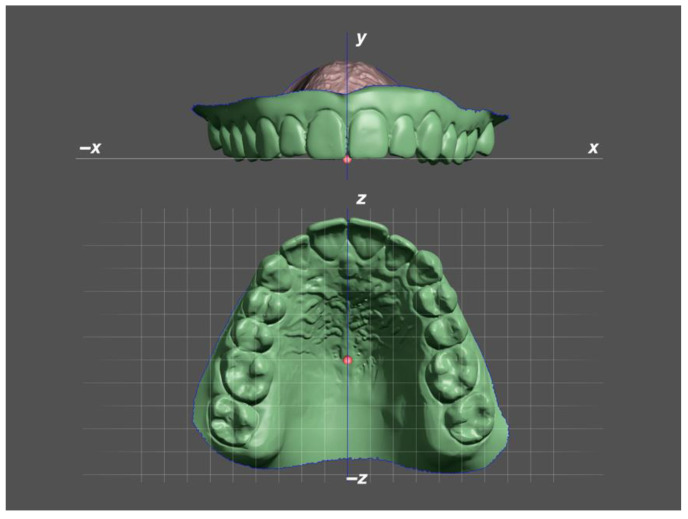
Alignment of the maxilla within Meshmixer’s world frame according to the occlusal plane and the palatal suture. (x) Lateromedial, (z) anteroposterior, (y) inferosuperior direction.

**Figure 2 jpm-12-01893-f002:**
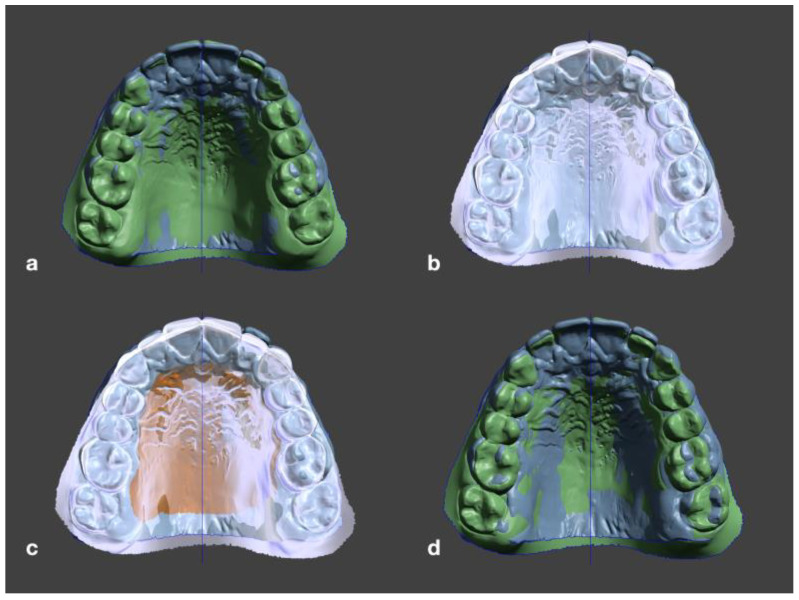
(**a**) Manual preregistering of the T_1_ STL to the T_0_ STL. (**b**) Selection of the T_0_ STL as the target object. (**c**) Manual selection of the palatal surface of T_1_ with the brush tool. (**d**) Automatic alignment of T_1_ to T_0_ with an error tolerance of 0.01 mm.

**Figure 3 jpm-12-01893-f003:**
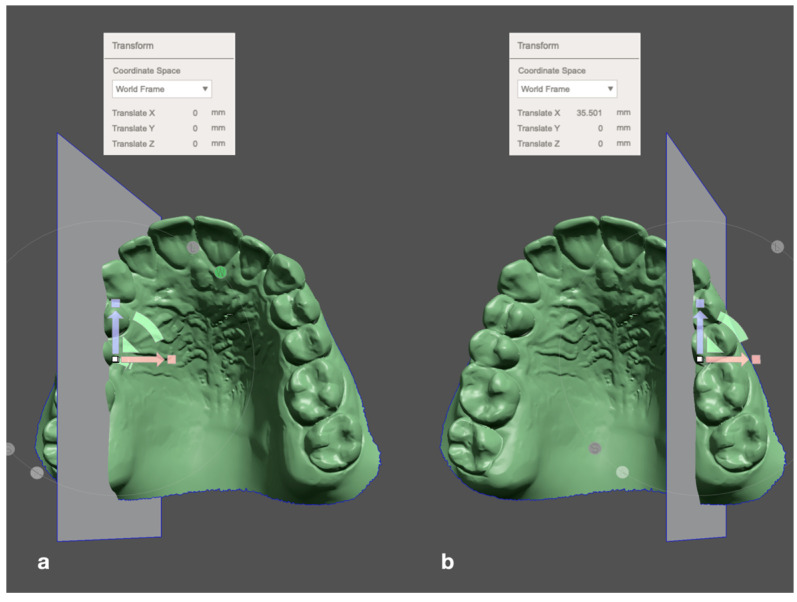
Example of transverse measurement (perspective view for illustration). (**a**) A surface object oriented parallel to the median plane was placed on the upper right canine landmark. (**b**) With the transform tool (colored arrows), the surface object was moved along the *x*-axis to the corresponding landmark at the upper left canine (W = world frame, L = local frame). The distance is displayed in the Transform window.

**Figure 4 jpm-12-01893-f004:**
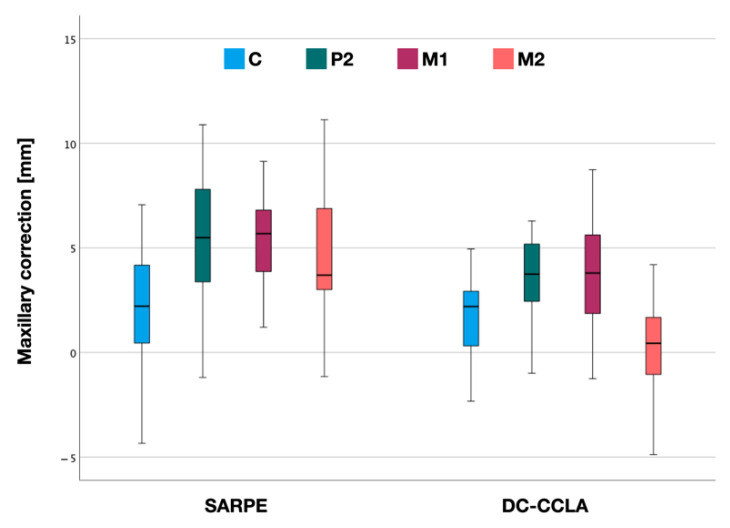
Maxillary correction after surgically assisted rapid palatal expansion (SARPE) and after dentoalveolar compensation (DC-CCLA) according to the distances in the canine (C), 2nd premolar (P2), 1st molar (M1) and 2nd molar (M2) region.

**Figure 5 jpm-12-01893-f005:**
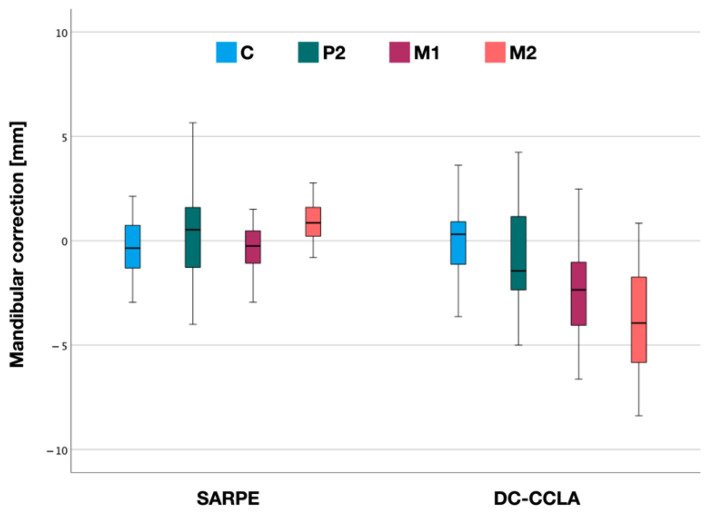
Mandibular correction after surgically assisted rapid palatal expansion (SARPE) and after dentoalveolar compensation (DC-CCLA) according to the distances in the canine (C), 2nd premolar (P2), 1st molar (M1) and 2nd molar (M2) region.

**Figure 6 jpm-12-01893-f006:**
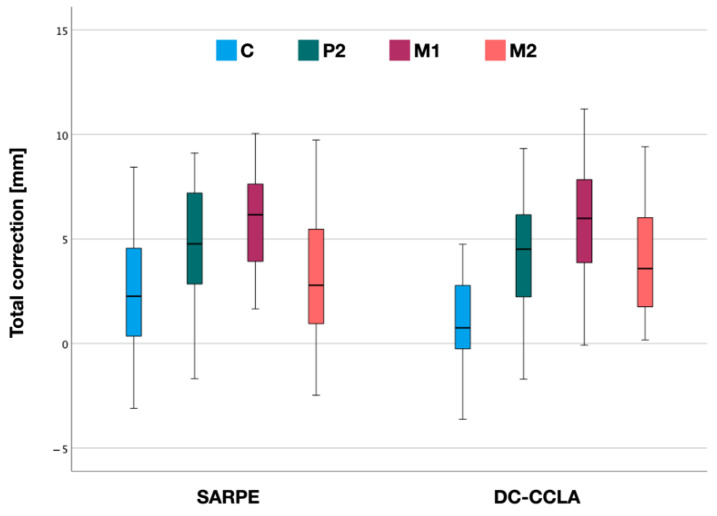
Total correction after surgically assisted rapid palatal expansion (SARPE) and after dentoalveolar expansion with CCLA according to the distances in the canine (C), 2nd premolar (P2), 1st molar (M1) and 2nd molar (M2) region.

**Figure 7 jpm-12-01893-f007:**
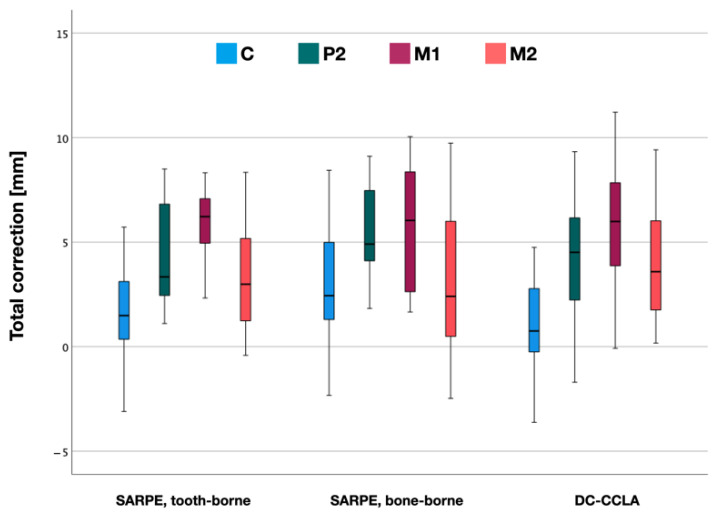
Total correction after surgically assisted rapid palatal expansion (SARPE) divided into tooth-borne, bone-borne appliances and DC-CCLA. Distances in the canine (C), 2nd premolar (P2), 1st molar (M1) and 2nd molar (M2) region.

**Figure 8 jpm-12-01893-f008:**
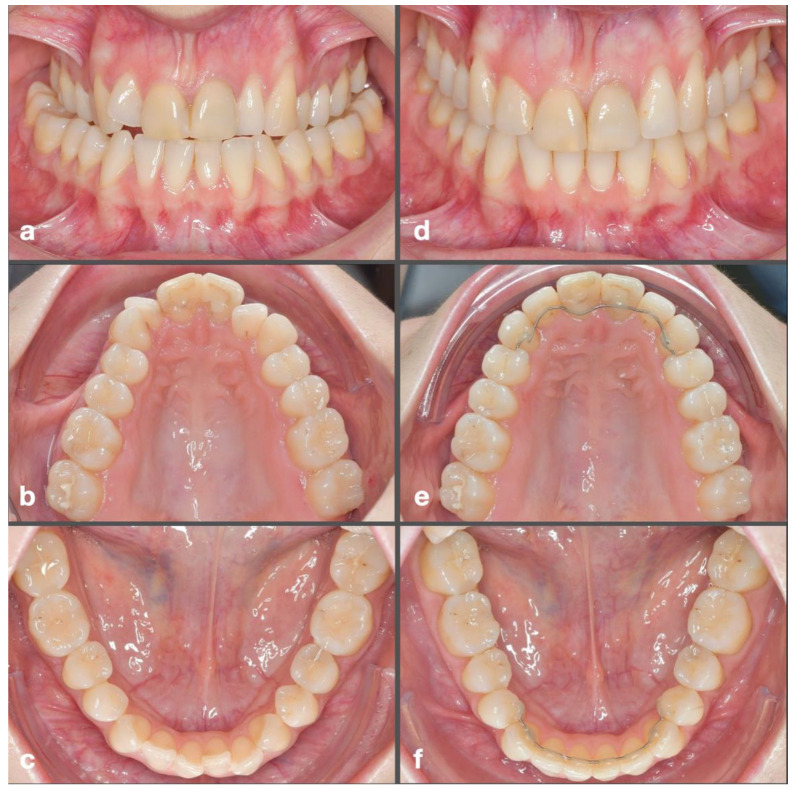
Clinical example of non-surgical dentoalveolar compensation of a posterior crossbite with completely customized lingual appliances. Situation before treatment with bilateral crossbite (**a**) due to a narrow upper (**b**) and broad lower arch (**c**). Post-treatment situation after crossbite correction (**d**) due to dentoalveolar expansion of the upper arch (**e**) and dentoalveolar constriction of the lower arch (**f**). Visually, no signs of overexpansion are apparent. Constriction of the lower arch increases from C to M2.

**Table 1 jpm-12-01893-t001:** Baseline characteristics (mean ± SD) of the groups. Intervention, gender, Angle Class and age in years.

	SARPE	DC-CCLA	*p*
Female	19 (44%)	25 (66%)	
age (years) at T_0_	26.7 ± 9.7	32.1 ± 12.3	0.086
age (years) at T_1_	30.1 ± 9.6	34.4 ± 12.4	0.255
Male	24 (56%)	13 (34%)	
age (years) at T_0_	27.8 ± 9.8	27.0 ± 14.2	0.404
age (years) at T_1_	30.9 ± 9.5	30.0 ± 14.4	0.276
Angle class I	1	20	
Angle class II	14	13	
Angle class III	28	5	

**Table 2 jpm-12-01893-t002:** Mean values (+ = expansion, − = constriction), standard deviations (sd) and 95% confidence intervals (CI), and minimum (min) and maximum (max) values for maxillary (Mx) correction, mandibular (Md) correction and total crossbite correction.

Group	Jaw	Tooth	Mean	SD	95% CI	Min	Max
SARPE	Mx	C	2.28	3.02	1.26–3.30	−4.34	11.11
		P2	5.40	2.98	4.39–6.41	−1.19	10.89
		M1	5.69	2.60	4.80–6.57	1.21	14.79
		M2	4.71	2.76	3.78–5.64	−1.15	11.13
	Md	C	−0.44	1.66	−1.05–0.17	−5.35	2.13
		P2	0.36	2.31	−0.49–1.21	−4.00	5.65
		M1	−0.36	1.52	−0.92–0.20	−3.51	3.19
		M2	0.86	1.20	0.42–1.30	−3.36	2.77
	total	C	2.21	2.62	1.23–3.18	−3.10	8.44
		P2	4.82	2.68	3.82–5.82	−1.68	9.11
		M1	5.82	2.35	4.94–6.70	1.66	10.05
		M2	3.26	3.05	2.12–4.40	−2.47	9.74
DC-CCLA	Mx	C	1.78	1.91	1.08–2.48	−2.33	4.95
		P2	4.07	2.85	3.02–5.11	−0.99	10.23
		M1	3.66	2.45	2.77–4.56	−1.25	8.74
		M2	0.36	2.64	−0.61–1.33	−6.59	6.43
	Md	C	0.20	1.67	−0.49–0.89	−3.63	3.62
		P2	−0.69	2.53	−1.73–0.36	−5.00	4.24
		M1	−2.70	2.26	−3.63–−1.77	−6.63	2.48
		M2	−3.54	2.64	−4.63–−2.45	−8.38	0.84
	total	C	1.29	2.10	0.36–2.22	−3.62	4.75
		P2	4.38	2.71	3.17–5.58	−1.70	9.33
		M1	6.00	2.78	4.77–7.23	−0.08	11.22
		M2	4.11	2.86	2.85–5.38	0.17	9.42

Note the similarity between the maximum values of total correction between the groups (grey background).

## Data Availability

The data underlying this article will be shared on reasonable request to the corresponding author.

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
