# Peer review of "Non-Surgical Transversal Dentoalveolar Compensation with Completely Customized Lingual Appliances versus Surgically Assisted Rapid Palatal Expansion in Adults—The Amount of Posterior Crossbite Correction"

_jpm, 2022, doi:10.3390/jpm12111893_

Round 1
Reviewer 1 Report
Dear authors,
This is a very worthwhile article. it is well written and I have commended to for publication. .
Thank you for submitting it to the journal.
Author Response
Dear Reviewer,
Thank you for taking the time to review our manuscript. Please find our answers in the attachment.
Sincerely,
Jonas Q. Schmid

Reviewer 2 Report
1. Introduccion
-the biggest problem when we make conservative maxillary expansion (I suggest to change it to “slow expansion” ) is the possibility to create fenestration o abscence of cortical bone…. You are referring an article (10) to defend that with the slow expansion we can have problems with the osseous articulations of the maxilla with the surrounding bones. And article 10 talks about SARPE
- But the evidence is weak and it remains a subjective decision to opt for surgically assisted rapid palatal expansion (SARPE) in late adolescents and early adults [13] because the midpalatal suture is not the only resistance to conservative maxillary expansion and the circummaxillary sutures have to be considered as well [14]. This leaves SARPE, segmental osteotomy and dentoalveolar compensation as possible therapy options in skeletally more mature patients.
But there is a lot of evidence that shows a parallel expansion pattern in late adolesce and ealy adults. Don´t affirm that the best option is SARPE o mandibular constriction
-To date there is no consensus about either the extent or procedure for SARPE [10,15] or the appliance used [16,17].
What do you mean with appliance used?
- The expansion pattern of bone-borne transpalatal distractors 55 seems to be more V-shaped [18] in comparison to a more parallel expansion with tooth 56 borne appliances [19].
The tooth bone-borne expander (MARPE) has shown a more parallel expansion. Don´t talk about distractors. Please revies more literature
- Dental effects of SARPE have been found to be greater 58 than skeletal effects. Therefore, SARPE can be considered primarily a molar expansion 59 procedure rather than a bodily skeletal expansion of the maxilla [21].
Please see more literature because of SARPE . Karlien Asscherickx , Elke Govaerts , Johan Aerts , Bart Vande Vannet . Maxillary changes with bone-borne surgically assisted rapid palatal expansion: A prospective study. Am J Orthod Dentofacial Orthop. . 2016 Mar;149(3):374-83. Shows just the opposite, that only 21% of the obtained expansion was dental. If we use bone borne expander with SARPE even less.
2. Material and Method
- criteria were adult patients with crossbite of two or more teeth in the 85 posterior segments. Exclusion criteria were patients with clefts. Teeth that were moved in 86 the sagittal direction for space closure because of extractions or space opening for 87 prosthetic replacement of a tooth were excluded from the measurement.
Please describe more de exclusion and inclusion criteria. Patients with agenesia, dental absence, anterior ortho treatment. This should be excluded
-You need to include a method to determine the transversal skeletal problem. Both groups should show similar results in maxillary constriction. This method should be done with CBCT if all patients have at the initial situation or with dental models.
-The brand of lingual brackets should be described. Cross bite elastics were used? This should be explained
- The decision for a bone-borne or tooth- 107 borne appliance was made by the referring orthodontist on a case-by-case basis.
Both types of expanders can be at the same group. This sample must be limited, using bone or dental expanders.
- In my opinion gingiva recession should be consider and measure T0 and T1 with both techniques. Also dental torque or inclinations should be consider
- I don´t understand the way of measuring. The figure 3 is not clear. It says a mark at the right canine and at the picture there is a white square landmark at the 1st premolar
- 3. Results
- 13 males (mean age 27.0 ± 14.2 years) the minimum age is 13 years, and this is not consider adulthood. This not comparable with the others group where the minimum age was 17 or 18 years old
-4. Discussion
-The null hypothesis that net 253 transversal changes with SARPE are greater than with dentoalveolar correction must be 254 rejected.
To be rejected in my opinion this article should provide evidence about dental inclination, buccal and lingual cortical width.
Author Response

(The authors gave the same response as above.)

Reviewer 3 Report
Dear Authors
The work is generally well written, but has some important shortcomings:
-It should be emphasized more clearly in the discussion that you are only assessing the transversal alignment of the teeth without assessing the change in their angular inclination. Of course, you wrote that one treatment method is dento-alveolar displacement and the other is orthopedic. But it is necessary to refer to all implications of this fact more extensively.
- You wrote that the measurement on the models was described as accurate, but since this is a key issue for this work, I would use the results of the meta-analysis assessing the accuracy and reliability of measurements on the scanned models in the transverse dimension. I would include such work in References instead of individual original studies.
- Conlusions need to be redrafted - it cannot be written that the dento-alveolar cotectomy is an equally effective method of cross-bite correction. This can be written when a new example of the long-term stability of such treatment is examined. One can only write about the clinical effectiveness of such a procedure, subject to severe limitations.
Sincerely Yours
Reviewer
Author Response

(The authors gave the same response as above.)

Round 2
Reviewer 2 Report
The comments are the article. The introductions and the discussion should be improved

Author Response
Dear Reviewer,
Thank you very much for taking the time to review our manuscript again. Please find our answers in the attachment.
Sincerely,
Jonas Q. Schmid
